materials science

microbiologically influenced corrosion, pH, sulfate-reducing bacteria, stainless steel

**Author for correspondence:**
K. Kannoorpatti
e-mail: krishnan.kannoorpatti@cdu.edu.au

This article has been edited by the Royal Society of Chemistry, including the commissioning, peer review process and editorial aspects up to the point of acceptance.

# Effect of pH regulation by sulfate-reducing bacteria on corrosion behaviour of duplex stainless steel 2205 in acidic artificial seawater

T. T. T. Tran[1], K. Kannoorpatti[1], A. Padovan[2] and S. Thennadil[1]

[1]Energy and Resources Institute, College of Engineering, Information Technology and Environment, and [2]Research Institute for Environment and Livelihoods, College of Engineering, Information Technology and Environment, Charles Darwin University, Darwin, Northern Territory 0909, Australia

TTTT, 0000-0003-4656-5018; KK, 0000-0003-0782-1525; ST, 0000-0001-9392-7857

Sulfate-reducing bacteria (SRB) can regulate environmental pH because of their metabolism. Because local acidification results in pitting corrosion, the potential capacity of pH regulation by SRB would have important consequences for electrochemical aspects of the bio-corrosion process. This study focused on identifying the effect of pH on the corrosion of duplex stainless steel 2205 in a nutrient-rich artificial seawater medium containing SRB species, *Desulfovibrio vulgaris*. Duplex stainless steel samples were exposed to the medium for 13 days at 37°C at pH ranging from 4.0 to 7.4. The open-circuit potential value, sulfide level, pH and number of bacteria in the medium were recorded daily. Electrochemical impedance spectroscopy (EIS) and potentiodynamic polarization were used to study the properties of the biofilms at the end of the experiments and the corrosion behaviour of the material. Inductively coupled plasma mass spectrometry was used to measure the concentration of cations Fe, Ni, Mo, Mn, Cr in the experimental solution after 13 days. Scanning electron microscopy and energy-dispersive X-ray spectroscopy (EDX) were used for surface analysis. The results showed the pH changed from acidic values set at the beginning of the experiment to approximately pH 7.5 after 5 days owing to bacterial metabolism. After 13 days, the highest iron concentration was in the solution that was initially at pH 4 accompanied by pitting on the stainless steel. Sulfur was present on all specimens but with more sulfur at pH 4 in the EDX spectra. EIS showed the film resistance of the specimen at pH 4 was much lower than

at pH 7.4 which suggests the corrosion resistance of the stainless steel was better at higher pH. The results of this study suggest that the corrosion process for the first few days exposure at low pH was driven by pH in solution rather than by bacteria. The increasing pH during the course of the experiment slowed down the corrosion process of materials originally at low pH. The nature and mechanism of SRB attack on duplex stainless steel at different acidic environments are discussed.

# 1. Introduction

Microbiologically influenced corrosion (MIC) is an electrochemical process of deteriorating material with the involvement of bacteria. It can increase corrosion rates up to 2–3 times compared to abiotic corrosion and has been the cause of severe corrosion problems in many industries [1]. Although there are numerous kinds of bacteria that can play an important role in the accelerated corrosion of materials, the chief culprits are the anaerobic sulfate-reducing bacteria (SRB). They catalyse sulfate from the environment to sulfide through their metabolism. SRB grow and corrode metals by electrochemical mechanisms through a series of oxidation (anodic) and reduction (cathodic) reactions of chemical species in direct contact with, or near, the metallic surface. There have been several studies which have showed that these microorganisms cause pitting on passive materials including stainless steel [2,3].

Previous research indicates that the pH of the environment may have a significant influence on SRB growth [1,4]. SRB can survive in acidic environments such as mine tailings and acid drainage owing to their ability to regulate pH to more favourable conditions [5,6]. The pH regulation ability of SRB has been used to remediate acidic areas such as the biogenic neutralization of acid rock drainage environments [7–9]. In abiotic environments, the corrosion rate of duplex stainless steel decreases with increased pH [10]. Thus, in environments containing SRB, the corrosion behaviour of materials may be different to abiotic environments.

The previous report has indicated that environmental pH has significant effect on the corrosion of materials not only in the abiotic environment [11–15], but also in the microbial environment [16–18]. However, the recent literature has focused mostly on MIC behaviour of duplex stainless steel at a neutral pH [19–22]. Corrosion on stainless steel caused by SRB in different pH environments has received less attention. The effect of pH on the corrosion rate of carbon steel in SRB medium was studied in previous literature [17]. However, parameters important in understanding MIC mechanisms were not recorded, such as the change in pH during the time and the growth of bacteria. Several previous studies show that some *Desulfovibrio* species activities are inhibited at pH below 5 while some other *Desulfovibrio* species have the ability to grow well even at very low pH environment [1,4]. In this study, the parameters including changes in open-circuit potential value (OCP), pH, bacteria concentration, and dissolved sulfide concentration were measured. The objective of this research was to conduct a comprehensive study on the corrosion behaviour of duplex stainless steel in different pH environments containing SRB.

# 2. Material and methods

## 2.1. Materials

Twelve duplex stainless steel 2205 (DSS 2205) coupons (10 mm × 10 mm × 2 mm) were used for the experiment at four different pH levels: pH 4, pH 5, pH 6 and pH 7.4. All coupons were mounted in a mould of non-conducting epoxy resin with an insulated copper wire to act as working electrodes. They were then polished to around 1 μm finish. After polishing, the coupons were rinsed with water, degreased with acetone, rinsed with distilled water, immersed in 80% ethanol for 2 h and finally dried in biohazard cabinet to prevent any bacterial contamination before the experiments. The chemical composition of DSS 2205 was determined through an energy-dispersive X-ray fluorescence spectrometer (XRF-8100) (%): Fe 66.318; Mn 1.678; S 0.053; V 0.108; Si 0.432; Cr 22.1; Ni 6.116; Cu 0.304; Mo 2.891. Four coupons per pH were immersed for 13 days for testing OCP, electrochemical impedance spectroscopy (EIS), sulfide level, pH level, enumerating bacteria, inductively coupled plasma mass spectrometry (ICPMS) and scanning electron microscope analysis (SEM). Another eight coupons were used for potential dynamic polarization scan: four samples per pH were immersed for 3 days and the rest were immersed for 13 days. The polarization curves were recorded at the end of exposure to the corrosion environment.

## 2.2. Medium and test conditions

Nutrient-rich artificial seawater was the chosen environment for the corrosion study. This consisted of modified Baar's medium (g l$^{-1}$): MgSO$_4$ 0.2; sodium citrate 0.5; CaSO$_4$ 0.1; NH$_4$Cl 0.1; K$_2$HPO$_4$ 0.05; sodium lactate 3.5; yeast extract 1.0 added to 1 l of artificial seawater prepared according to ASTM 114-98 (g l$^{-1}$) [23]: NaCl 24.53; MgCl$_2$ 5.2; Na$_2$SO$_4$ 4.09; CaCl$_2$ 1.16; KCl 0.695; NaHCO$_3$ 0.201; KBr 0.101; H$_3$BO$_3$ 0.027; SrCl$_2$ 0.0025, NaF 0.003 and high pure water.

The test medium was distributed to four 500 ml sterile glass bottles with 400 ml each. The pH of the solution in each bottle was adjusted to pH 4, 5, 6 and 7.4 using 1 M hydrochloric acid and 1 M sodium hydroxide. The test medium was purged with nitrogen gas for 1 h and sterilized by autoclaving for 15 min at 121°C. pH 7.4 was chosen as it is the optimum condition for pure SRB culture to grow in modified Baar's medium and for comparison [17,24,25].

*Desulfovibrio vulgaris* (ATCC 7757) (In Vitro Technologies, VIC) was retrieved from −80°C glycerol stock and cultured in 500 ml modified Baar's medium for 48 h at 37°C under anaerobic conditions. After approximately 48 h, 10 ml of bacteria culture medium was removed for determining the bacterial concentration. The bacterial cells were harvested by centrifugation (5000 rpm, 10 min) and resuspended in 10 ml of high pure water, stained with 0.4% trypan blue and counted using a haemocytometer. Five ml of culture medium was added to each 500 ml glass bottle containing nutrient-rich artificial seawater to give a final bacterial concentration of approximately $3.17 \times 10^4$ cells ml$^{-1}$.

Additionally, glutaraldehyde 2.5% was prepared for staining bacterial biofilm before doing surface analyses and phosphate-buffered saline 1X (PBS) was prepared for samples preparation before doing surface analysis.

The experiment was carried out for 13 days at 37°C which falls within the optimum temperature range for the growth of mesophilic bacteria.

In this paper, all the solution samples changed to pH 7.5 so the term pH 4, pH 5, pH 6, pH 7.4 refers to the initial pH of the samples.

## 2.3. Analytical methods

The analytical methods which were used for the study includes measuring OCP, EIS, potentiodynamic polarization, sulfide and pH level, enumerating bacteria, ICPMS analysis and surface analysis.

## 2.4. Open-circuit potential

All electrochemical experiments were performed in a three-electrode cell. A platinum-coated electrode was used as a counter electrode, an Ag/AgCl electrode as a reference electrode and the working electrode (the specimens in other words) was introduced face up to allow bacteria to settle on the surface. A nitrogen gas layer was added to the top of the cell to create fully anaerobic conditions inside the cell. The electrochemical experiments were performed using VERSASTAT3-300 potentiostat and the results analysed using VERSASTUDIO software. OCP value of each specimen was recorded daily.

## 2.5. Electrochemical impedance spectroscopy

EIS was recorded after 3 days and 13 days of exposure to the corrosive medium. EIS has been used to study MIC and its biofilm formation and its interaction with the material surface [26]. The tests were carried out at OCP and the amplitude value was 20 mV with frequency range from 0.05 to 100 000 Hz. The impedance data were analysed by an equivalent circuit using software ZSIMPWIN which was integrated with VERSASTUDIO.

## 2.6. Potentiodynamic polarization

The polarization curves were recorded potentiodynamically using a scan rate of 0.5 mV s$^{-1}$ and starting from −0.25 V versus OCP to transpassive potential after 3 days and 13 days exposure. The corrosion potential ($E_{corr}$) and corrosion current density ($I_{corr}$) were obtained from the polarization curves.

## 2.7. Sulfide level and pH level

A 1 mm diameter hole was made on the three-electrode cell and covered with epoxy for maintaining airtight conditions. A sample of the medium was removed from each bottle daily using a sterile

syringe with a needle through the epoxy hole. After taking the samples, the hole was then covered by a new epoxy layer to maintain anaerobic conditions. Sulfide levels were measured using a Hach DR300 Colorimeter. In addition, the pH of each sample was measured using a pH meter. Each measurement was done three times.

## 2.8. Enumerating bacteria

Bacteria were enumerated daily from a 0.5 ml portion of test solution by counting cells under light microscopy using a haemocytometer with trypan blue staining agent. Total cells including dead and live cells were counted. This measurement was repeated three times.

## 2.9. Inductively coupled plasma mass spectrometry analysis

On the final day of the experiment, 15 ml solution from each bottle was filtered through a 0.45 µm filter to remove bacteria. The solutions were then analysed for metal concentrations in an Agilent 7500ce ICPMS which is an octopole reaction system using a standard addition calibration method for seawater. The reporting limit for Cr, Mn, Fe, Ni and Mo are 0.05, 0.10, 0.30, 0.10, 0.30 ppb, respectively. This experiment was done three times.

## 2.10. Surface analysis

All the working electrodes were removed from the solution after 13 days, rinsed three times with 1X PBS, fixed with 2.5% glutaraldehyde in PBS for 30 min, then washed twice with high purity water. The specimens were dehydrated in an ethanol series (25%, 50%, 75%, 90% and 100%) for 10 min each and then dried in a biohazard cabinet prior to SEM-EDX analysis.

## 2.11. Quality assurance and control

All the experiments were conducted in the University molecular laboratory which is a certified PC2 laboratory which undergoes yearly inspection by the Institutional Biosafety Committee. The ICPMS tests are calibrated in the laboratory that follows documented analytical protocols based on United States Environmental Protection Agency methodology and other published methodology and includes extensive Quality Control analyses with every sample batch. The laboratory subscribes to QUASIMEME (Quality Assurance of Information for Marine Environmental Monitoring in Europe) which is a subscription-based international quality assurance programme for marine environmental monitoring, and the laboratory has participated actively since 1997. Laboratory performance studies include four rounds per annum of 'blind' analysis of seawater, sediment and biota samples provided by QUASIMEME. Submitted data are statistically assessed to provide an external quality assurance for chemical measurements in the marine environment. The laboratory's performance in QUASIMEME is consistently of very high quality. The potentiostat was calibrated using internal and external dummy cells provided by the instrument manufacturer. The pH meter was calibrated using standard buffer solutions. The Hach DR 3000 Colorimeter was calibrated using standard samples and reagents provided by Hach. The JEOL SEM-energy-dispersive X-ray spectroscopy (EDX) was calibrated and serviced regularly by a JEOL technician.

# 3. Results

All error bars in figures 2–5 represent ± one standard deviation of the measurements determined using three repeat measurements.

## 3.1. Open-circuit potential values

Figure 1 shows the OCP values which declined rapidly in all four pH experiments, reaching a minimum within about 5 days followed by a stable region. The OCP values in the pH 7.4 experiment declined more rapidly than the pH 4 experiment. OCP indicates the corrosion state of the iron; the thicker the biofilm, the lower the OCP values because of difficulties of ion transfer through the biofilm.

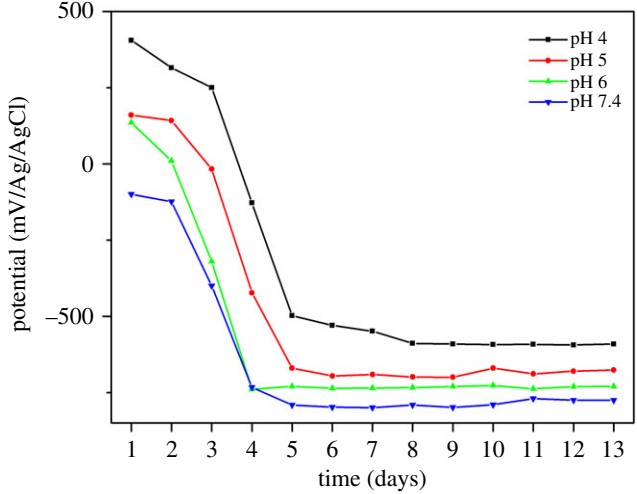

**Figure 1.** OCP values of DSS 2205 coupons for the duration of the experiment at different starting pH.

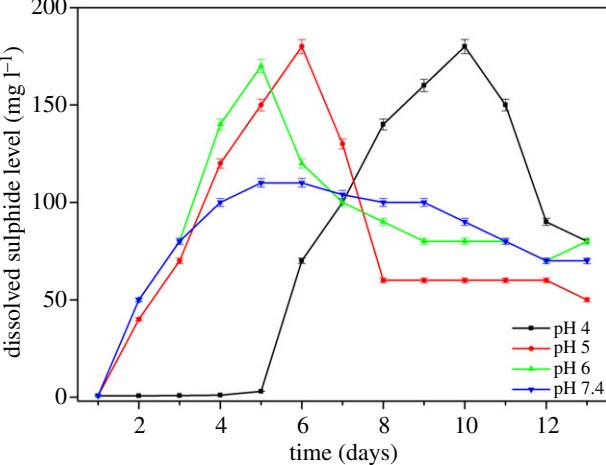

**Figure 2.** Dissolved sulfide level versus time data for samples with different starting pH.

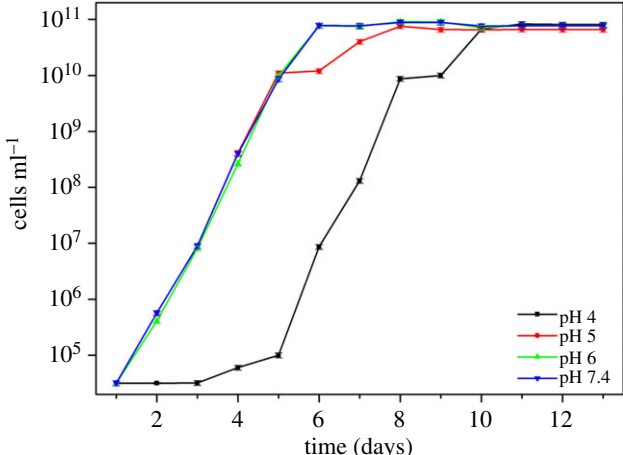

**Figure 3.** Changes in the concentration of SRB at each pH.

## 3.2. Sulfide level

Figure 2 shows the dissolved sulfide levels over the course of the experiment. There were sharp increases of dissolved sulfide levels at pH 5 and 6 environments to a maximum of approximately 175 mg l$^{-1}$ before

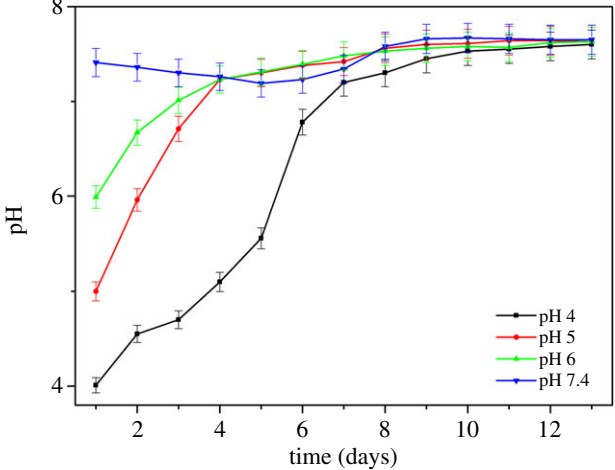

**Figure 4.** Variation of pH during exposure time for different starting pH values.

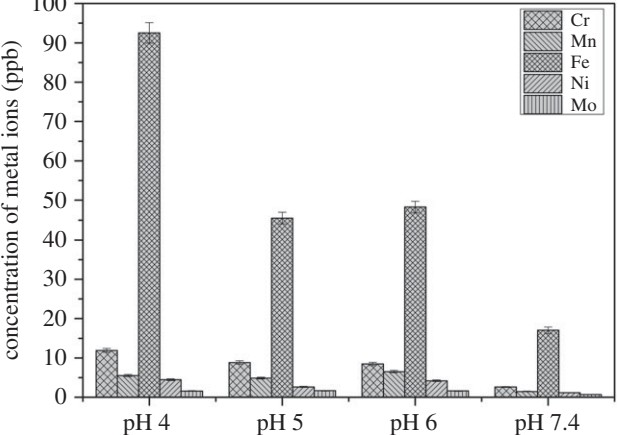

**Figure 5.** Concentration of metal ion in different pH solutions.

decreasing around day 5. At pH 4, sulfide levels also increased to approximately 175 mg l$^{-1}$ but after a 5-day lag. At pH 7.4, sulfide levels rapidly increased but only to approximately 100 mg l$^{-1}$ and then gradually decreased for the remainder of the experiment. It is interesting to note that the maximum for the pH 7.4 sample was about half of the maximum of other samples which would imply lower sulfate reducing activities for pH 7.4 samples.

### 3.3. Bacteria concentrations and pH level

Figures 3 and 4 show the bacteria concentration and pH of the media during the course of the experiment, respectively. The concentration of SRB at pH 5, 6 and 7.4 increased rapidly around the first 6 days and then remained stable. At pH 4, there was only a gradual increase in bacteria concentration for the first 5 days, after which the concentration rose sharply to similar levels as the other pH experiments. This could imply that for the first 5 days, a portion of the bacterial population died or cells were shocked and unable to function properly owing to low pH, and after 5 days, when the pH of the environment reached a value conducive to the bacteria, they started growing rapidly. This is in good agreement with the sulfide level result. Figure 2 shows that, for pH 4, the sulfide production was nearly absent during the first few days.

The pH of all solutions irrespective of starting pH reached 7.5 after 7–10 days.

### 3.4. Concentrations of metal ions

Figure 5 shows after 13 days, the concentration of Cr, Mn, Fe, Ni and Mo ions in the pH 4 environment was higher than at other pH values. In particular, the concentration of Fe ions in the pH 4 solution was more than 4 times that compared to the pH 7.4 solution, indicating a poor corrosion resistance at low pH.

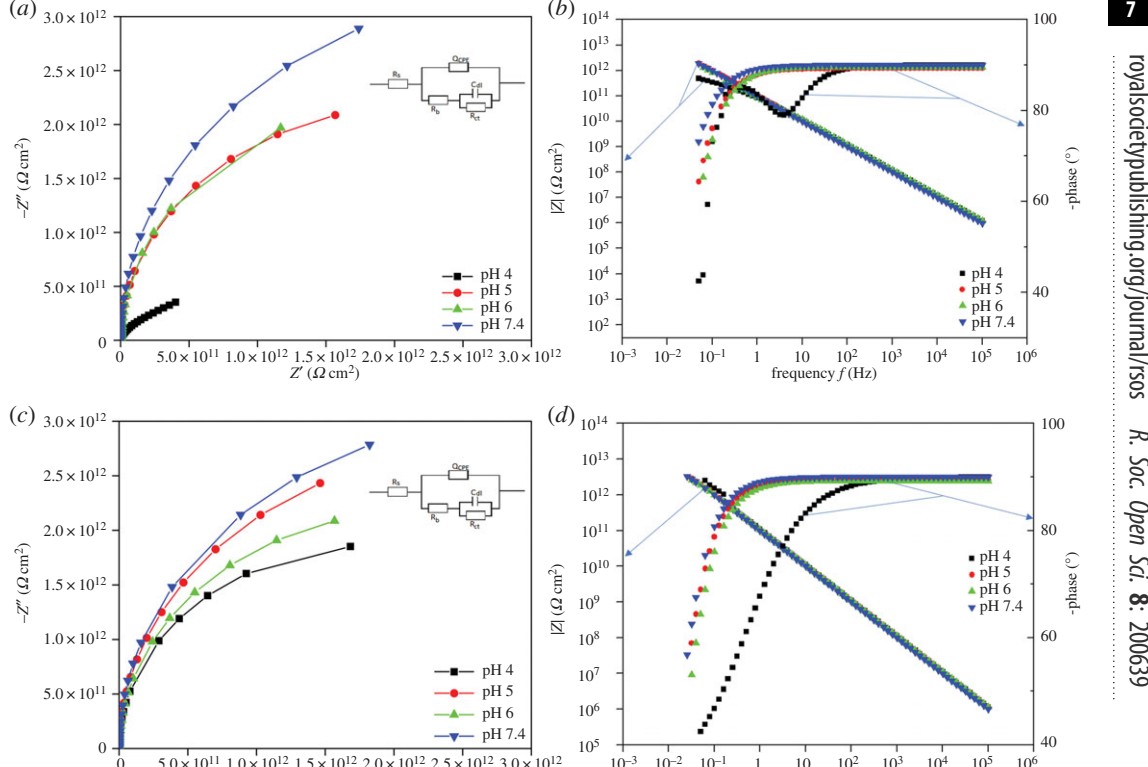

**Figure 6.** Nyquist plot and corresponding Bode plot of each specimen after 3 day's exposure (a,b), and 13 day's exposure (c,d), based on the equivalent electric circuit model given in (a,c).

## 3.5. Electrochemical impedance spectroscopy studies

Figure 6 shows the Nyquist plot and Bode plot of specimens in different pH environments after 3 days and after 13 days with its electrical equivalent circuit (EEC). A constant phase element (CPE) is introduced to the model as it represents a deviation from a true capacitive behaviour. CPE usually substitutes the capacitance in electrical circuits because of the inhomogeneous conditions (e.g. electrode roughness, coating and distribution of reaction rate) [27]. It represents the deviation from true capacitive behaviour. CPE usually substitutes the capacitance in electric circuits because of the inhomogeneous conditions (e.g. electrode roughness, coating, and distribution of reaction rate). Its admittance and impedance were defined by the following equations, respectively:

$$Y_{CPE} = Y_0(j\omega)^n \tag{3.1}$$

$$Z_{CPE} = \frac{1}{Y_0(j\omega)^n}, \tag{3.2}$$

where $Y_0$ is the magnitude of the CPE, $j$ is the imaginary number, $\omega$ is the angular frequency and $n$ is the CPE power index ($n < 1$).

In the EEC model, $R_b$ is the resistance of passive film/biofilm formed on the specimen surface, $R_{ct}$ is the charge transfer resistance, $C_{dl}$ is the capacitance of the electrical double layer and $Q_{CPE}$ is the CPE parameter. The spectra for all specimen in different pH environment fitted well to the EEC model.

## 3.6. Potential dynamic polarization studies

Figure 7 presents the polarization curves of samples after 3 days and 13 days exposure. There are no obvious differences in the shape of the curves which suggests different pH does not have a significant effect on the kinetics of the corrosion process. The corrosion current density and corrosion potential are presented in table 1. After 3 days, corrosion of the sample in pH 4 had the highest current density which reveals the highest corrosion rate compared to other samples. However, after 13 days immersion, the current density of the sample in the pH 4 environment just slightly increased (0.026 µA).

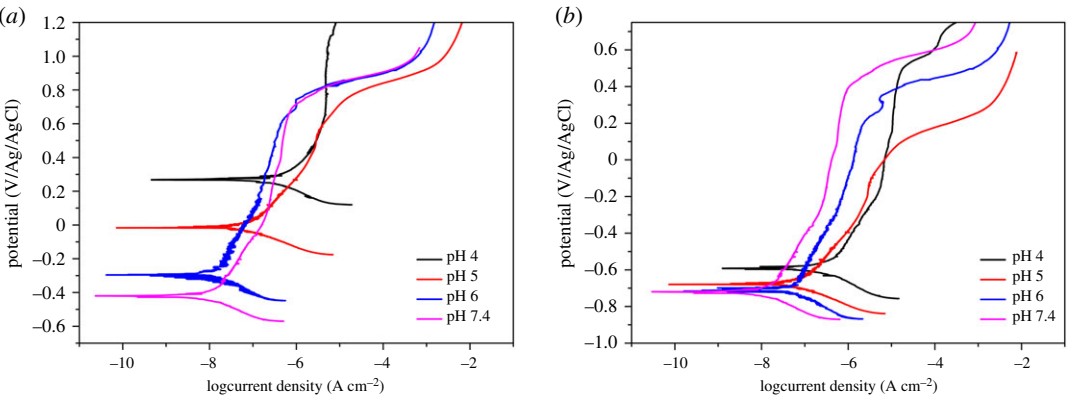

**Figure 7.** Potentiodynamic polarization curve of samples in different pH environment after 3 days (*a*) and 13 days (*b*) exposure.

**Table 1.** Corrosion current densities and corrosion potentials of samples.

|  | 3 days exposure | | 13 days exposure | |
| --- | --- | --- | --- | --- |
|  | $I_{corr}$ (µA cm$^{-2}$) | $E_{corr}$ (mV versus Ag/AgCl) | $I_{corr}$ (µA cm$^{-2}$) | $E_{corr}$ (mV versus Ag/AgCl) |
| pH 4 | 0.498 | 251 | 0.524 | −592 |
| pH 5 | 0.086 | −17 | 0.087 | −680 |
| pH 6 | 0.041 | −320 | 0.050 | −732 |
| pH 7.4 | 0.030 | −402 | 0.039 | −779 |

## 3.7. Scanning electron microscopy–energy dispersive X-ray spectroscopy

SEM images showed the presence of microbes on the surface and, in some cases, pitting could be clearly seen on the surface of the metals (figure 8*a*–*d*). EDX analysis (figure 8*e*–*h*) was performed at 200× in order to achieve an optimum level of counts. The magnification of SEM images was higher to show clearly pitting and biofilm. The spectra of all the DSS 2205 coupons showed a peak of sulfur. The specimen from the pH 4 environment had the highest concentration of sulfur while the specimen from pH 7.4 had the lowest concentration of sulfur. There was no oxygen detected in any of the specimens. Therefore, the detected sulfur element was most likely in sulfide form.

# 4. Discussion

## 4.1. The pH regulation ability of sulfate-reducing bacteria

This study showed that SRB can regulate pH, changing the pH from acidic conditions to approximately pH 7.5 after 5 days (figure 4). SRB consumes protons from the environment resulting in raising the pH in the immediate microenvironment around the cells to a pH which supports bacterial survival [28]. Sulfate reduction is a proton-consuming process [29] thus, this process is favoured with decreasing pH. The sulfate reduction rate depends on the proton concentration in the environment. At lower pH or higher proton concentration, the sulfate reduction rate will be higher. This could explain the total dissolved sulfide in pH 5, 6 and 7.4 solution samples (figure 2). The sulfate reduction rates at pH 5 and 6 were higher than at pH 7.4, thus resulting in higher dissolved sulfide. Once the SRB gradually consume protons, the pH of the bulk environment would increase.

The bacterial concentration increased at pH 5, 6 and 7.4 during the first 4 days. This was not the case at pH 4 which could indicate that at lower pH, a significant number of cells either died or were shocked and could not function or replicate during this time. At pH 4, the proton concentration is high so the sulfate reduction rate necessary to maintain an adequate microenvironment around the cells was

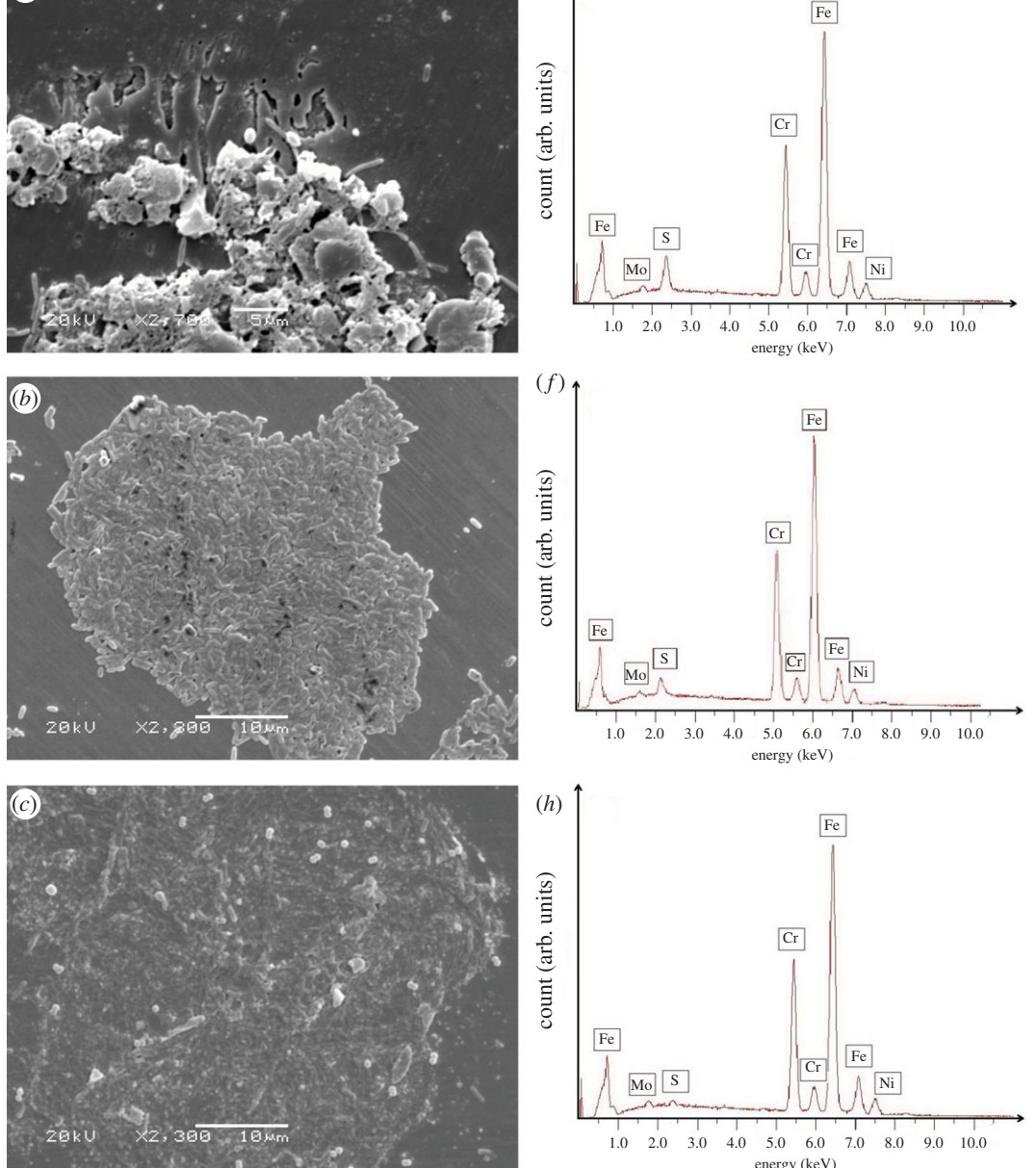

**Figure 8.** SEM images of specimens in different pH samples: (*a*) at pH 4, (*b*) at pH 5, (*c*) at pH 6, (*d*) at pH 7.4 and EDX spectra of specimens in different pH samples: (*e*) at pH 4, (*f*) at pH 5, (*g*) at pH 6, (*h*) at pH 7.4.

higher than the sulfate reduction rate the bacteria could afford. Thus, this resulted in the death or inactivation of part of the bacterial population (figure 3). After about 5 days, when the pH increased to 5 the total bacteria cells increased rapidly (figure 3), and sulfide reduction also increased sharply (figure 2).

## 4.2. Microbial corrosion behaviour of duplex stainless steel during the exposure time

SRB is found everywhere in many different environments predominantly in oxygen-free environments. In anaerobic conditions, the alternative cathodic reaction to hydrogen evolution, such as oxidation by gaseous or dissolved oxygen, is not available either. Therefore, the main cathodic reaction is the dissociation of water to form hydrogen ions. The bacteria mediate the anaerobic reduction of $SO_4^{2-}$ and hydrogen as an electron acceptor to produce S and/or $H_2S$ via half-cell electrode reactions (equations (4.1)–(4.4)) [30]. The corresponding Nernst equation can provide a chemical state of sulfur over a range of pH and potential. The anodic reaction is the oxidation of metals such as Fe to $Fe^{2+}$.

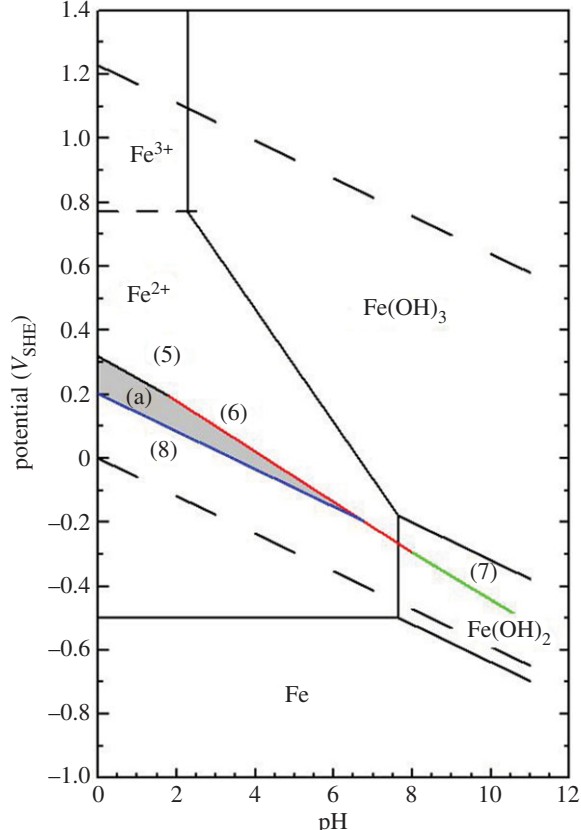

**Figure 9.** Pourbaix diagram for Fe with superimposed sulfate reducing process.

The sulfides react with the metal ions and form metal sulfides which precipitate on the metal surface around the SRB and mixed with the biofilm [31]:

$$HSO_4^- + 7H^+ + 6e^- = S + 4H_2O \quad E = 0.339 - 0.069\,pH + 0.01\log[HSO_4^-], \tag{4.1}$$

$$SO_4^{2-} + 8H^+ + 6e^- = S + 4H_2O \quad E = 0.357 - 0.079\,pH + 0.01\log[SO_4^{2-}], \tag{4.2}$$

$$SO_4^{2-} + 10H^+ + 8e^- = H_2S + 4H_2O \quad E = 0.311 - 0.074\,pH + 0.007\log[SO_4^{2-}]/P_{H_2S}, \tag{4.3}$$

$$S + 2H^+ + 2e^- = H_2S \quad E = 0.142 - 0.059\,pH - 0.03\log[H_2S]. \tag{4.4}$$

Figure 9 shows the Pourbaix diagram of iron and sulfate reduction activities [30]. There is a narrow potential region 'a' where sulfur is stable among reactions (4.1) to (4.4). From figure 1, it can be seen that the potentials of the specimen in pH 4, which was 405 mV versus Ag/AgCl for the first few days, were in the region 'a' (in figure 5) where sulfur, $S^0$, is stable. Sulfur is very corrosive to stainless steel [32] because it will catalyse anodic dissolution and hinder passivation [33]. This may have been responsible for the high dissolution of iron at the end of the experiments in pH 4 (figure 5). Then, when the potential value dropped to a lower value around −500 mV versus Ag/AgCl, reaction (4.4) occurred and produced $S^{2-}$ which could combine with dissolved metal ion to form metal sulfide on the surface of specimens which were shown by EDX results (figure 8). This could be also seen in EIS results (figure 6), the total impedance of specimen in pH 4 environment was much lower than other specimens after 3 days exposure. After 13 days, the impedance increased owing to the formation of metal sulfide and thick biofilm on the surface of the material. This also happened to specimens in pH 5 and 6 environments, however, at a lower rate. At pH 7.4, the potential value (figure 1) and the presence of little sulfur in EDX results (figure 8) produced low corrosion rate represented by dissolved iron and other elements (figure 5).

Based on the Nyquist plot (figure 4), the diameter of the curve of the coupon at pH 7.4 after 3 days was the highest indicating its impedance was the highest, whereas the diameter of the coupon at pH 4 was the lowest. This suggests that the corrosion resistance of the sample at pH 4 was much lower than in the other samples, in good agreement with the polarization curves (figure 7). After 13 days the impedance of the coupon at pH 4 increased considerably but was lower than other specimens.

This could indicate the formation of a film of corrosion and/or scale products and biofilm with greater thickness, thus, substantiating the increase in impedance values. For coupons at pH 5, 6 and 7.4, the impedance showed a slight decrease after 13 days exposure compared to after 3 days exposure. This could indicate that the passive layer becomes slightly defective owing to the formation of minor quantities of metal sulfide as a corrosion product. EDX results also showed the presence of sulfur element on the surface of specimens which might be in the form of metal sulfide. In other words, all the specimens showed their corrosion tendency within 13 days exposure.

The Bode plot (figure 6b,d) showed the total impedance of all specimens was high at low frequencies. For coupons at pH 4 after 3 days exposure, the total impedance at the low frequency was lower than other specimens which would suggest that the thickness of biofilm at pH 4 was much lower than the thickness of specimens at higher pH.

Regarding the phase curves in the Bode plot (figure 6b,d), after 3 days exposure, the phase angle spectra (figure 6b) of specimens in pH 5, 6 and 7.4 environments had one-time constant at low frequency. When the frequency increased, the phase angle increased and remained at 90°. This would indicate the stability of the film formation including biofilm and passive film on the surface. However, the spectrum obtained from the specimen in the pH 4 environment showed a very different response owing to the apparent formation of two relaxation times. This would probably indicate 2-stage phenomena in the corrosion mechanism of the specimen at pH 4 after 3 days exposure which might be the process of proton transfer to the surface of the material through the biofilm which did not occur with specimens at pH 5, 6 and 7.4. This suggests that at pH 4, a very high concentration of proton results in high penetration of protons through the biofilm to the surface of the specimen for the first 3 days and could result in dissolving the passive film and corroding the specimen.

After 13 days exposure, the phase angle spectra (figure 6d) of the specimen in the pH 4 environment was found to have one-time constant which might be the result of decrease in proton transfer to the surface of the material. In addition, the total impedance of this specimen increased after 13 days exposure. This is possibly owing to SRB activities in regulating the pH environment within a few days and resulted in decreased proton transfer to the surface of the material which should result in less corrosion of the material after 13 days exposure compared to 3 days exposure. Thus, corrosion was greater in the first few days of exposure. Potentiodynamic polarization results showed good agreement with this result. For the first 3 days, the corrosion current density of the sample in pH 4 was the highest compared to other samples, however, after 13 days immersion, this sample's current density just slightly increased by 5%. These results reveal for the first few day's exposure, corrosion of samples was mainly driven by the pH of the solution. As the low pH experiment proceeded, the pH changed to 7.5 which resulted in slowing down of the corrosion process of the samples. In other words, it was shown that the bacteria are able to change the pH of the environment. The increase in pH and the increase in bacterial concentration resulted in thick biofilm formation on the surface of the coupons which acted like a barrier [34], lowering the corrosion rate of the sample at initial pH 4.

Some *Desulfovibrio* species cannot grow at pH below 5 owing to inhibition of sulfate reduction [35,36]. However, there are other *Desulfovibrio* species that grow well at very low pH (e.g. 2.9) [28,37–39]. This study showed that *D. vulgaris* can grow at low pH (e.g. 4) after an initial lag phase. Metabolism and sulfate reduction were not limited as the total sulfide produced by bacteria at pH 4 was similar to other pH environments. The corrosion rate of DSS 2205 was shown to be greater at low pH than optimized pH for cultivating bacteria (pH 7.4). This finding was similar to previous reports on carbon steels at pH from 5 to 7 with the same bacterial species [17]. At pH 4, the corrosion process was dominated by the electrochemical process rather than by microbial corrosion for the first few day's exposure. This also occurred with *Pseudomonas aeruginosa* in a previous study on carbon steel [18]. After the initial lag phase, SRB grew quickly and formed a biofilm on the material surface acting like barrier to reduce the corrosion process.

There are studies that have shown that the biofilm and biogenic layers of corrosion products can protect materials [34,40,41] by improving the adherence of the passive film to the metal. By contrast, other studies have found the opposite [42–44] with increased corrosion owing to heterogeneities at the metal surface [45]. If the passive film is disrupted, then the passive film can be repaired if there is enough oxygen in the fluid. Fluid flow is generally good for materials that produce passivity due to the availability of oxygen. However, in the case of SRB environments, there is very little oxygen, and this can make stainless steels vulnerable to corrosion. Biofilms can be protecting under such cases. The experiments in this research were conducted in a static environment. It is possible that in flow conditions where the speed of the fluid is high, the biofilm may be disrupted, and the corrosion behaviour of the metal might be different. Efforts are underway by the authors to study the effect of fluid flow on microbial corrosion.

# 5. Conclusion

The effect of pH on corrosion behaviour of duplex stainless steel in acidic seawater environment was evaluated by observing pH level, dissolved sulfide level, OCP, bacteria population, EIS, potentiodynamic polarization, SEM-EDX and ICPMS. The main conclusions obtained from this work are presented below:

— The results showed the environmental pH was changed by SRB after around 5 days.
— The highest iron concentration was at pH 4 and this was 3 times higher than at pH 7.4 indicating increased release of iron due to corrosion at lower pH.
— EIS results showed the film resistance of the specimen at pH 4 was much lower than at pH 7.4 which suggests the corrosion resistance of the duplex stainless steel was better at pH 7.4 than at pH 4.
— Corrosion current density was higher at first time immersion and reduced with time of exposure due to SRB activities. This suggests that SRB pH regulation activities could slow down the corrosion processes of duplex stainless steel in very low pH environments.

Data accessibility. The raw data are uploaded to the Dryad Digital Repository: https://doi.org/10.5061/dryad.3ffbg79fh [46].

Authors' contributions. T.T.T.T. conceived, designed the work, analysed data and wrote the manuscript. K.K., A.P. and S.T. assisted in designing the work, analysing data and revised the manuscript. All authors read and approved the final manuscript to be published.

Competing interests. We declare we have no competing interests

Funding. T.T.T.T.'s PhD candidature is funded by Charles Darwin University RTP scholarship funding.

Acknowledgements. T.T.T.T. is grateful for the internship at INPEX Technical Research Centre, Japan where the XRF analyses were performed. Dr Nam Nguyen Dang is thanked for helpful advice.

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
