## [Reviewer comments · Royal Society Open Science]

Review History

RSOS-200639.R0 (Original submission)

Review form: Reviewer 1

Is the manuscript scientifically sound in its present form?

Yes

Are the interpretations and conclusions justified by the results?

Yes

Is the language acceptable?

Yes

Do you have any ethical concerns with this paper?

No

Have you any concerns about statistical analyses in this paper?

No

Recommendation?

Accept with minor revision (please list in comments)

Comments to the Author(s)

This paper is focusing on identifying the effect of pH on the corrosion of duplex stainless steel 2205 in a nutrient rich artificial seawater medium containing Sulphate reducing bacteria species, namely *Desulfovibrio vulgaris*. The manuscript contains original results and deserves to be published after minor revision. Here are my comments and observations:

1. Page 3, last line: The title "Analytical methods" does not reflect the content of the section. Maybe something is missing...
2. Page 4, lines 49-52: I suggest moving the phrase "In this paper, all the solution samples changed to pH 7.5 so the term pH 4, pH 5, pH 6, pH 7.4 refers to the initial pH of the samples..." to the previous section where pH level is discussed.
3. Figure 3. The parameter on the x axis should be "Time" expressed in days, like in Figure 1, 2 and 4.
4. Page 5, line 27. "QCPE is the CPE parameter" is not clear. Q is the pseudo-capacitance corresponding to CPE.
5. Page 6. For clarity reasons, I suggest writing the chemical reactions in the same line with the corresponding expressions of e. (The notation "E" is maybe better than "e")
6. Table 1. E should be expressed in mV vs. Ag/AgCl electrode; I_{corr} should be expressed with the same number of decimals (3 are sufficient).
7. Figure 5. The error bars are not visible; please write on the x-axis "pH" instead "ph"
8. Figure 6. The impedance spectra should be orthonormal (same values on both axes). If the lines represent the curves simulated with the equivalent electrical circuit, this should be mentioned in the legend of the figure.
9. The conclusions should be more detailed.

Review form: Reviewer 2

Is the manuscript scientifically sound in its present form?

Yes

Are the interpretations and conclusions justified by the results?

Yes

Is the language acceptable?

Yes

Do you have any ethical concerns with this paper?

Yes

Have you any concerns about statistical analyses in this paper?

Yes

Recommendation?

Major revision is needed (please make suggestions in comments)

Comments to the Author(s)

It seems to be a meaningful result of MIC research, I would like you to consider the following some points.

Although the SRB used in this study seems to be a popular corrosive strain, the experimental results hardly mentioned the corrosive effect caused by this strain, but rather emphasized the corrosion inhibitory effect.

Based on the above, ①In the experiments using the same strain, there are some results showing the reproduction of MIC even in neutral pH culture conditions. What is the difference between these results and yours? ②In order to discuss the effects of corrosion under acidic conditions, it should be preferable to set the same pH abiotic experimental condition as a control. Could you tell me the reason why?

I would like you to clarify your views and position regarding these.

In addition, clarify the novelty and purpose of research more than in the introduction

At the same time, I would like it reflected in the title of the paper.

Based on these, I would like to request that the research novelty and purpose be made clearer in the introduction part. Similarly, research content should be reflected more clearly in the title of the paper.

Some items pointed out are specifically described below.

- (1) Some keywords such as SRB or anaerobic should be included in the title.
- (2) P.1, l.11: Please clarify what "film" means.
- (3) P.1, l.11~12: The sentence starting with "was" is incomplete.
- (4) P1, l.41: Does Sulfide mean iron sulfide?
- (5) P1, l.48: In this study, while the suppression of corrosion by biofilm produced by SRB is discussed, there is few mentions about the corrosion phenomenon assisted by SRB. Usually it is easy to find the corrosion phenomenon assisted by SRB (may be same strain) and occurred in neutral pH condition, in some literature. Please see also (8).
- (6) P.3 l.8~9: What is the difference between corrosion under condition A and corrosion under abiotic environment? So what is the contribution of SRB to corrosion in this case?
- (7) P.4, l.31: Please indicate (%) unit. Whether is mass fraction or atomic fraction?
- (8) P.4, l.51: The strain ATCC 7757 has been treated as a popular corrosive bacterium in many bio-corrosion studies. However, your studies emphasized the corrosive effect of pH on the material, rather than the bacterial corrosive ability. Please explain your view on these.
- (9) P.5, l.56: Did you measure biofilm thickness?
- (10) P.9, Figure 1, 2, P.10, Figure 4, 5 : The letters, legend, and plot are small and unclear.
- (11) P.11. Figure 6: The circuit model is unclear and unreadable.
- (12) P12, Figure 8: (1) For the observation and analysis by SEM/EDX, did you perform coating process (i.e. vapor deposition) on a sample to avoid charge-up?
(2) Did you performed EDS analysis on the entire field of view (P.4 l.45)?
In that case, how did you consider the effect of magnification (Figure 8) ?
(3) In the EDX results, there is no information about the label and scale in the vertical axis.
- (13) P.5, l.58: The bacterial concentration increased
You mentioned that both the number of live cells and dead cells were evaluated, (p.4, l.32) but I could not find the results of identifying both.
Is the bacterial count concentration shown in Figure 3 mean only the number of viable bacteria?
- (14) P.6, l.24: (below) Figure 1
Please read the numerical values in the figure and explain them in a little more detail and more carefully.
- (15) P.6, l.29: What does (S4) mean?

Decision letter (RSOS-200639.R0)

Dear Dr Tran Thi Thuy:

Title: Effect of pH regulation by microbes on corrosion behaviour of duplex stainless steel 2205 in acidic artificial seawater environment
Manuscript ID: RSOS-200639

The editor assigned to your manuscript has now received comments from reviewers. We would like you to revise your paper in accordance with the referee and Subject Editor suggestions which can be found below (not including confidential reports to the Editor). Please note this decision does not guarantee eventual acceptance.

Please submit your revised paper before 05-Jul-2020. Please note that the revision deadline will expire at 00.00am on this date. If we do not hear from you within this time then it will be assumed that the paper has been withdrawn. In exceptional circumstances, extensions may be possible if agreed with the Editorial Office in advance. We do not allow multiple rounds of revision so we urge you to make every effort to fully address all of the comments at this stage. If deemed necessary by the Editors, your manuscript will be sent back to one or more of the original reviewers for assessment. If the original reviewers are not available we may invite new reviewers.

On behalf of the Subject Editor Professor Anthony Stace and the Associate Editor Dr Nadia Martinez Villegas.

RSC Associate Editor:

Comments to the Author:

Quality assurance and control must be clearly mentioned in the Materials and Methods section.

RSC Subject Editor:

Comments to the Author:

(There are no comments.)

Reviewers' Comments to Author:

Reviewer: 1

Comments to the Author(s)

This paper is focusing on identifying the effect of pH on the corrosion of duplex stainless steel 2205 in a nutrient rich artificial seawater medium containing Sulphate reducing bacteria species, namely *Desulfovibrio vulgaris*. The manuscript contains original results and deserves to be published after minor revision. Here are my comments and observations:

1. Page 3, last line: The title "Analytical methods" does not reflect the content of the section. Maybe something is missing...
2. Page 4, lines 49-52: I suggest moving the phrase "In this paper, all the solution samples changed to pH 7.5 so the term pH 4, pH 5, pH 6, pH 7.4 refers to the initial pH of the samples..." to the previous section where pH level is discussed.
3. Figure 3. The parameter on the x axis should be "Time" expressed in days, like in Figure 1, 2 and 4.
4. Page 5, line 27. "QCPE is the CPE parameter" is not clear. Q is the pseudo-capacitance corresponding to CPE.
5. Page 6. For clarity reasons, I suggest writing the chemical reactions in the same line with the corresponding expressions of e. (The notation "E" is maybe better than "e")
6. Table 1. E should be expressed in mV vs. Ag/AgCl electrode; Icorr should be expressed with the same number of decimals (3 are sufficient).
7. Figure 5. The error bars are not visible; please write on the x-axis "pH" instead "ph"
8. Figure 6. The impedance spectra should be orthonormal (same values on both axes). If the lines represent the curves simulated with the equivalent electrical circuit, this should be mentioned in the legend of the figure.
9. The conclusions should be more detailed.

Reviewer: 2

Comments to the Author(s)

It seems to be a meaningful result of MIC research, I would like you to consider the following some points.

Although the SRB used in this study seems to be a popular corrosive strain, the experimental results hardly mentioned the corrosive effect caused by this strain, but rather emphasized the corrosion inhibitory effect.

Based on the above, ①In the experiments using the same strain, there are some results showing the reproduction of MIC even in neutral pH culture conditions. What is the difference between these results and yours? ②In order to discuss the effects of corrosion under acidic conditions, it should be preferable to set the same pH abiotic experimental condition as a control. Could you tell me the reason why?

I would like you to clarify your views and position regarding these.

In addition, clarify the novelty and purpose of research more than in the introduction

At the same time, I would like it reflected in the title of the paper.

Based on these, I would like to request that the research novelty and purpose be made clearer in the introduction part. Similarly, research content should be reflected more clearly in the title of the paper.

Some items pointed out are specifically described below.

- (1) Some keywords such as SRB or anaerobic should be included in the title.
- (2) P.1, l.11: Please clarify what "film" means.
- (3) P.1, l.11~12: The sentence starting with "was" is incomplete.
- (4) P1, l.41: Does Sulfide mean iron sulfide?
- (5) P1, l.48: In this study, while the suppression of corrosion by biofilm produced by SRB is discussed, there is few mentions about the corrosion phenomenon assisted by SRB. Usually it is easy to find the corrosion phenomenon assisted by SRB (may be same strain) and occurred in neutral pH condition, in some literature. Please see also (8).
- (6) P.3 l.8~9: What is the difference between corrosion under condition A and corrosion under abiotic environment? So what is the contribution of SRB to corrosion in this case?
- (7) P.4, l.31: Please indicate (%) unit. Whether is mass fraction or atomic fraction?
- (8) P.4, l.51: The strain ATCC 7757 has been treated as a popular corrosive bacterium in many bio-corrosion studies. However, your studies emphasized the corrosive effect of pH on the material, rather than the bacterial corrosive ability. Please explain your view on these.
- (9) P.5, l.56: Did you measure biofilm thickness?
- (10) P.9, Figure 1, 2, P.10, Figure 4, 5 : The letters, legend, and plot are small and unclear.
- (11) P.11. Figure 6: The circuit model is unclear and unreadable.
- (12) P12, Figure 8: (1) For the observation and analysis by SEM/EDX, did you perform coating process (i.e. vapor deposition) on a sample to avoid charge-up?
(2) Did you performed EDS analysis on the entire field of view (P.4 l.45)
In that case, how did you consider the effect of magnification (Figure 8) ?
(3) In the EDX results, there is no information about the label and scale in the vertical axis.
- (13) P.5, l.58: The bacterial concentration increased
You mentioned that both the number of live cells and dead cells were evaluated, (p.4, l.32) but I could not find the results of identifying both.
Is the bacterial count concentration shown in Figure 3 mean only the number of viable bacteria?
- (14) P.6, l.24: (below) Figure 1
Please read the numerical values in the figure and explain them in a little more detail and more carefully.
- (15) P.6, l.29: What does (S4) mean?

Author's Response to Decision Letter for (RSOS-200639.R0)

See Appendix A.

RSOS-200639.R1 (Revision)

Review form: Reviewer 1

Is the manuscript scientifically sound in its present form?

Yes

Are the interpretations and conclusions justified by the results?

Yes

Is the language acceptable?

Yes

Do you have any ethical concerns with this paper?

No

Have you any concerns about statistical analyses in this paper?

No

Recommendation?

Accept as is

Comments to the Author(s)

The answers of the authors to the reviewers' comments are satisfactory. Consequently, I recommend the publication of the manuscript in its present form.

Review form: Reviewer 2

Is the manuscript scientifically sound in its present form?

Yes

Are the interpretations and conclusions justified by the results?

Yes

Is the language acceptable?

Yes

Do you have any ethical concerns with this paper?

No

Have you any concerns about statistical analyses in this paper?

Yes

Recommendation?

Accept with minor revision (please list in comments)

Comments to the Author(s)

This paper is an important contribution and I recommend that it be accepted for publication.

However, the numbering notation in Figure 8 is small and difficult to decipher. I would like to request a clearer letter.

Decision letter (RSOS-200639.R1)

Dear Dr Tran Thi Thuy:

Title: Effect of pH regulation by sulphate reducing bacteria on corrosion behaviour of duplex stainless steel 2205 in acidic artificial seawater
Manuscript ID: RSOS-200639.R1

It is a pleasure to accept your manuscript in its current form for publication in Royal Society Open Science. The chemistry content of Royal Society Open Science is published in collaboration with the Royal Society of Chemistry.

On behalf of the Subject Editor Professor Anthony Stace and the Associate Editor Dr Nadia Martinez Villegas.

RSC Associate Editor:

Comments to the Author:

Your paper has been recommended for publication. However, please be aware that Reviewer 2 has made a minor but important comment that must be attended before during proofreading.

RSC Subject Editor:

Comments to the Author:

(There are no comments.)

Reviewer(s)' Comments to Author:

Reviewer: 1

Comments to the Author(s)

The answers of the authors to the reviewers' comments are satisfactory. Consequently, I recommend the publication of the manuscript in its present form.

Reviewer: 2

Comments to the Author(s)

This paper is an important contribution and I recommend that it be accepted for publication. However, the numbering notation in Figure 8 is small and difficult to decipher. I would like to request a clearer letter.

Appendix A

Response to reviewers' and editor's comments

RSC Associate Editor:

We would like to thank you for your constructive comments on the manuscript. We have made changes based on your comments as indicated below in our responses to your comments:

Quality assurance and control must be clearly mentioned in the Materials and Methods section.

Response: A new section has been added to the Materials and Methods section.

Quality Assurance and Control

All the experiments were conducted in the University molecular lab is a certified PC2 lab which undergoes yearly inspection by the Institutional Biosafety Committee. The ICPMS tests are calibrated in the lab that follows documented analytical protocols based on USEPA methodology and other published methodology and includes extensive Quality Control analyses with every sample batch. The lab subscribes to QUASIMEME (Quality Assurance of Information for Marine Environmental Monitoring in Europe) which is a subscription based international quality assurance program for marine environmental monitoring, and the lab participates actively since 1997. Laboratory performance studies include four rounds per annum of 'blind' analysis of seawater, sediment and biota samples provided by QUASIMEME. Submitted data is statistically assessed to provide an external quality assurance for chemical measurements in the marine environment. The lab's performance in QUASIMEME is consistently of very high quality. The potentiostat was calibrated using internal and external dummy cells provided by the instrument manufacturer. The pH meter was calibrated using standard buffer solutions. Hach DR 3000 Colorimeter was calibrated using standard samples and reagents provided by Hach. The JEOL SEM-EDX was calibrated and serviced regularly by a JEOL technician.

Reviewer 1:

We would like to thank you for your constructive comments on the manuscript. We have made changes based on your comments as indicated below in our responses to your comments:

1. Page 3, last line: The title “Analytical methods” does not reflect the content of the section. Maybe something is missing...

Response: I have changed the content of the referred section in the manuscript to:

“The analytical methods which were used for the study includes measuring open circuit potential (OCP), electrochemical impedance spectroscopy (EIS), potentiodynamic polarization, Sulphide and pH level, enumerating bacteria, Inductively coupled plasma mass spectrometry (ICPMS) analysis and surface analysis.”

We also have moved the paragraph which was previously in the Analytical methods section to the section “Medium and test conditions” final paragraph as that paragraph does not reflect the Analytical methods section.

2. Page 4, lines 49-52: I suggest moving the phrase “In this paper, all the solution samples changed to pH 7.5 so the term pH 4, pH 5, pH 6, pH 7.4 refers to the initial pH of the samples...” to the previous section where pH level is discussed.

Response: The mentioned paragraph has been moved to the section “Medium and test conditions” in the final paragraphs.

3. Figure 3. The parameter on the x axis should be “Time” expressed in days, like in Figure 1, 2 and 4.

Response: We have changed the parameter on the x axis and resized all the figures to make them clearer.

4. Page 5, line 27. “QCPE is the CPE parameter” is not clear. Q is the pseudo-capacitance corresponding to CPE.

Response: We have added more details about the parameter Q_{CPE} in the corresponding section 4 “Results”, part “Electrochemical impedance spectroscopy studies” as shown below:

The term constant phase element (CPE) was introduced to the model. It presents the deviation from true capacitive behaviour. CPE usually substitutes the capacitance in ECs because of the inhomogeneous conditions (e.g. electrode roughness, coating, and distribution of reaction rate). Its admittance and impedance were defined by following equations respectively:

$$Y_{CPE} = Y_0(j\omega)^n \quad (1)$$

$$Z_{CPE} = \frac{1}{Y_0(j\omega)^n} \quad (2)$$

Where Y_0 is the magnitude of the CPE, j is the imaginary number, ω is the angular frequency and n is CPE power index ($n < 1$).

5. Page 6. For clarity reasons, I suggest writing the chemical reactions in the same line with the corresponding expressions of e. (The notation “E” is maybe better than “e”)

Response: The notation has been changed from “e” to “E” in the manuscript as shown below:

6. Table 1. E should be expressed in mV vs. Ag/AgCl electrode; I_{corr} should be expressed with the same number of decimals (3 are sufficient).

Response: This has been corrected in the manuscript as shown below:

	3 days exposure		13 days exposure	
	I _{corr} (μA.cm ⁻²)	E _{corr} (mV vs Ag/AgCl)	I _{corr} (μA.cm ⁻²)	E _{corr} (mV vs Ag/AgCl)
pH 4	0.498	251	0.524	-592
pH 5	0.086	-17	0.087	-680
pH 6	0.041	-320	0.050	-732
pH 7.4	0.030	-402	0.039	-779

7. Figure 5. The error bars are not visible; please write on the x-axis “pH” instead “ph”

Response: The size of the figure has been increased to make the error bars visible. We have also corrected the term “ph” in Fig. 5.

8. Figure 6. The impedance spectra should be orthonormal (same values on both axes). If the lines represent the curves simulated with the equivalent electrical circuit, this should be mentioned in the legend of the figure.

Response: We have changed both axes to the same values and mentioned the equivalent electrical circuit in the caption.

9. The conclusions should be more detailed.

Response: More details have been added in the conclusions section as shown below:

“The effect of pH on corrosion behaviour of duplex stainless steel in acidic seawater environment was evaluated by observing pH level, dissolved sulphide level, OCP, bacteria population, EIS, potentiodynamic polarization, SEM-EDX and ICPMS. The main conclusions obtained from this work are presented below:

- The results showed the environmental pH was changed by SRB after around 5 days
- The highest iron concentration was at pH 4 and this was 3 times higher than at pH 7.4 indicating increased release of iron due to corrosion at lower pH
- EIS results showed the film resistance of the specimen at pH 4 was much lower than at pH 7.4 which suggests the corrosion resistance of the duplex stainless steel was better at pH 7.4 than at pH 4.
- Corrosion current density was higher at first time immersion and reduced with time of exposure due to SRB activities. This suggests that SRB pH regulation activities could slow down the corrosion processes of duplex stainless steel in very low pH environments.”

Reviewer 2:

We would like to thank you for your constructive comments on the manuscript. We have made changes based on your comments as indicated below in our responses to your comments.

It seems to be a meaningful result of MIC research, I would like you to consider the following some points.

Although the SRB used in this study seems to be a popular corrosive strain, the experimental results hardly mentioned the corrosive effect caused by this strain, but rather emphasized the corrosion inhibitory effect.

Response: The SRB species, *Desulfovibrio vulgaris*, was chosen because is a well-known corrosive strain. Our study's focus was on the impact of starting pH conditions on the MIC and thus the overall effect on corrosion is considered including inhibitory effect and pitting corrosion.

Based on the above, ① In the experiments using the same strain, there are some results showing the reproduction of MIC even in neutral pH culture conditions. What is the difference between these results and yours?

Response: Our results support previous results in neutral pH condition [1-4] and we have discussed the neutral pH condition in the context of the culture in low pH environment. We have made this clearer in the introduction at final paragraph as shown below:

“Previous report has indicated that environmental pH has significant effect on corrosion of materials not only in abiotic environment [11-15], but also in microbial environment [16-18]. However, the recent literature has focussed mostly on MIC behaviour of duplex stainless steel at a neutral pH [19-22]. Corrosion on stainless steel caused by SRB in different pH environments has received less attention. The effect of pH on the corrosion rate of carbon steel in SRB medium was studied in previous literature [17]. But parameters important in understanding MIC mechanisms were not recorded, such as the change in pH during time and the growth of bacteria. Several previous studies show that some *Desulfovibrio* species activities are inhibited at pH below 5 while some other *Desulfovibrio* species have ability to grow well even at very low pH environment [1, 4]. In this study, the parameters including changes in OCP, pH, bacteria concentration, dissolved sulphide concentration were measured. The objective of this research was to conduct a comprehensive study on the corrosion behaviour of duplex stainless steel in different pH environments containing SRB.”

② In order to discuss the effects of corrosion under acidic conditions, it should be preferable to set the same pH abiotic experimental condition as a control. Could you tell me the reason why?

Response: There are two reasons for not conducting the experiments under abiotic conditions. Firstly, the study emphasises the comparison of the corrosion behaviour between acidic environments and thus relative differences between them can be analysed without the need for abiotic experiments. Secondly, there are several previous studies on corrosion of stainless steel in abiotic environment under acidic condition which can be used for discussing the results of this study. In the manuscript, we have referred to these studies[5-9].

I would like you to clarify your views and position regarding these.

In addition, clarify the novelty and purpose of research more than in the introduction

At the same time, I would like it reflected in the title of the paper.

Based on these, I would like to request that the research novelty and purpose be made clearer in the introduction part.

Response: We have added sentences in the introduction to make the research novelty and purpose clearer in the final paragraph of Introduction section as shown above in ①.

Similarly, research content should be reflected more clearly in the title of the paper. Some items pointed out are specifically described below.

(1) Some keywords such as SRB or anaerobic should be included in the title.

Response: The title of the manuscript has been changed to: "Effect of pH regulation by sulphate reducing bacteria on corrosion behaviour of duplex stainless steel 2205 in acidic artificial seawater".

(2) P.1, I.11: Please clarify what "film" means.

Response: The term "film" can include biofilm formed on surface of material, passive film of stainless steel and corrosion products layer. In this study, we focus more on the biofilm rather than others. I have changed the term "films" to "biofilm" to remove the ambiguity.

(3) P.1, I.11~12: The sentence starting with "was" is incomplete.

Response: The dot "." before "was" was placed wrongly. The sentence has been rewritten.

(4) P.1, I.41: Does Sulphide mean iron sulfide?

Response: The term sulphide was used incorrectly. The word "sulphide" in the manuscript refers to sulphur element which was present in the EDS spectra of all samples. Therefore, "sulphide" has been changed to "sulphur".

(5) P.1, I.48: In this study, while the suppression of corrosion by biofilm produced by SRB is discussed, there is few mentions about the corrosion phenomenon assisted by SRB. Usually it is easy to find the corrosion phenomenon assisted by SRB (may be same strain) and occurred in neutral pH condition, in some literature. Please see also (8). (8) P.4, I.51: The strain ATCC 7757 has been treated as a popular corrosive bacterium in many bio-corrosion studies. However, your studies emphasized the corrosive effect of pH on the material, rather than the bacterial corrosive ability. Please explain your view on these.

Response: In this study, we studied the combined impact of pH of the environment and SRB rather than impact of pure SRB on corrosion behaviour of duplex stainless steel. The corrosive ability of SRB mostly depends on their metabolism including hydrogenase process (which depends on the type of SRB strain) which lead to cathodic depolarisation and their biogenic sulphide produced via metabolism [10]. In the discussion section, we focused more about sulphide produced by SRB which is corrosive to stainless steel materials and discussed it using the Pourbaix diagram (Fig. 9).

(6) P.3 I.8~9: What is the difference between corrosion under condition A and corrosion under abiotic environment? So what is the contribution of SRB to corrosion in this case?

Response: In this study, due to the presence of SRB, pH of the environment was increased to around neutral pH. This leads to the decrease of protons which is very corrosive to stainless steel in the chloride environment. This results in decrease in corrosion rate of materials as shown in the manuscript. Therefore, SRB might act as corrosion inhibitor as they increase environmental pH to neutral and reduced corrosion rate compared to abiotic environment for short time exposure.

(7) P.4, I.31: Please indicate (%) unit. Whether is mass fraction or atomic fraction?

Response: The portion is 0.5 mL which is not a fraction. So, every day, 0.5 mL of test solution was taken out to do bacterial enumeration.

(9) P.5, I.56: Did you measure biofilm thickness?

Response: We did not measure the biofilm thickness. Instead, the impedances of the samples were measured by EIS which can then be used to compare the biofilm thickness between samples. This method has been used in several previous studies [11, 12].

(10) P.9, Figure 1, 2, P.10, Figure 4, 5 : The letters, legend, and plot are small and unclear.

Response: We have increased the size of them and made them clearer.

(11) P.11. Figure 6: The circuit model is unclear and unreadable.

Response: We have increased its size to make it clearer.

(12) P12, Figure 8: (1) For the observation and analysis by SEM/EDX, did you perform coating process (i.e. vapor deposition) on a sample to avoid charge-up?

Response: We did not coat the samples because it might affect the EDX results. In fact, the sample is conductive, therefore there was no need to coat it.

(2) Did you performed EDS analysis on the entire field of view (P.4 I.45)? In that case, how did you consider the effect of magnification (Figure 8) ?

Response: The following has been added to the section on Results, part Surface analysis for more information:

“SEM images showed the presence of microbes on the surface and, in some cases, pitting could be clearly seen on the surface of the metals (Fig.8 a-d). EDX analysis (Fig.8 e-h) was performed at 200x in order achieve an optimum level of counts. The magnification of SEM images was higher to show clearly pitting and biofilm.”

Therefore, the magnification in SEM images do not have effect on result of EDX spectra.

(3) In the EDX results, there is no information about the label and scale in the vertical axis.

Response: I have added the label in the vertical axis. The spectra provide only qualitative information.

(13) P.5, I.58: The bacterial concentration increased

You mentioned that both the number of live cells and dead cells were evaluated, (p.4, I.32) but I could not find the results of identifying both.

Is the bacterial count concentration shown in Figure 3 mean only the number of viable bacteria?

Response: The cell concentration shown in Fig. 3 was the total cells i.e. viable and dead cells. SRB is obligate anaerobic bacteria and therefore it is vulnerable when exposed to open air environment. It is difficult to maintain all viable cells alive during the counting process under microscope under open air condition. Therefore, we chose to count the total cells.

(14) P.6, I.24: (below) Figure 1 ■ ■ ■ ■

Please read the numerical values in the figure and explain them in a little more detail and more carefully.

Response: We have added more details in this part in the manuscript to make it clearer to understand as shown below:

“Fig. 9 shows the Pourbaix-diagram of iron and sulphate reduction activities [30]. There is a narrow potential region ‘a’ where sulphur is stable among reactions (3) to (6). From Fig. 1 it can be seen that the potentials of specimen in pH 4, which was 405 mV vs Ag/AgCl for the first few days, were in the

region 'a' (in Fig. 5) where sulphur is stable. Sulphur is very corrosive to stainless steel [32] because it will catalyse anodic dissolution and hinder passivation [33]. This may have been responsible for high dissolution of iron at the end of the experiments in pH 4 (Fig. 5). Then, when the potential value dropped to a lower value around -500 mV vs Ag/AgCl, reaction (6) occurred and produced S^{2-} which could combine with dissolved metal ion to form metal sulphide on the surface of specimens which were shown by EDX results (Fig. 8). This could be also seen in EIS results (Fig. 6), the total impedance of specimen in pH 4 environment was much lower than other specimens after 3 days exposure. After 13 days, the impedance increased due to the formation of metal sulphide and thick biofilm on the surface of material. This also happened to specimens in pH 5 and 6 environment however at a lower rate. At pH 7.4, the potential value (Fig. 1) and the presence of little sulphur in EDX results (Fig. 8) produced low corrosion rate represented by dissolved iron and other elements (Fig. 5)."

(15) P.6, I.29: What does (S4) mean?

Response: It is the reaction (6). We have corrected it so it does not confuse the readers.

References used in this document

1. Dec, W., et al., *Corrosion behaviour of 2205 duplex stainless steel in pure cultures of sulphate reducing bacteria: SEM studies, electrochemical characterisation and biochemical analyses*. 2018. **69**(1): p. 53-62.
2. Jogdeo, P., et al., *Onset of microbial influenced corrosion (MIC) in stainless steel exposed to mixed species biofilms from equatorial seawater*. 2017. **164**(9): p. C532-C538.
3. Song, W., et al., *Microbial Corrosion of 2205 Duplex Stainless Steel in Oilfield-Produced Water*. 2018. **13**: p. 675-689.
4. Zhao, J., et al., *Investigation on mechanical, corrosion resistance and antibacterial properties of Cu-bearing 2205 duplex stainless steel by solution treatment*. 2016. **6**(114): p. 112738-112747.
5. Tang, D.Z., et al., *Effect of pH value on corrosion of carbon steel under an applied alternating current*. 2015. **66**(12): p. 1467-1479.
6. Malik, H.J.A.-C.m. and materials, *Effect of pH on the corrosion inhibition of mild steel in CO₂ saturated brine solution*. 2000. **47**(2): p. 88-93.
7. Wang, J., et al., *Effect of pH on corrosion behavior of 316L stainless steel in hydrogenated high temperature water*. 2018. **69**(5): p. 580-589.
8. Samosir, R. and S.L. Simanjuntak. *The influence of concentration and pH on corrosion rate in stainless steels–316 solution HNO₃ medium*. in *IOP Conference Series: Materials Science and Engineering*. 2017. IOP Publishing.
9. Cheng, X.Q., et al. *The influence of pH values on corrosion properties of 316L stainless steel in simulated circulating cooling water*. in *Advanced Materials Research*. 2011. Trans Tech Publ.
10. Enning, D. and J. Garrelfs, *Corrosion of Iron by Sulfate-Reducing Bacteria: New Views of an Old Problem*. 2014. **80**(4): p. 1226-1236.
11. Mansfeld, F. and B.J.C.S. Little, *A technical review of electrochemical techniques applied to microbiologically influenced corrosion*. 1991. **32**(3): p. 247-272.
12. Al-Kaseasbeh, Q.A., *Electrochemical Investigation of Corrosion Resistance of Weldments in Steel Bridges*. North Dakota State University, 2018.
13. Friel, J.J., C.E.J.M. Lyman, and Microanalysis, *Tutorial review: X-ray mapping in electron-beam instruments*. 2006. **12**(1): p. 2-25.